# Effect of enhanced peer PrEP referral with HIV self-testing delivery among young Kenyan women: A randomized controlled trial of peer networks

Katrina F. Ortblad[1,2]*, Njeri Wairimu[3], Carlos Culquichicon[4], Irene Njeru[3], Rachel C. Malen[1], Adriana M. Reedy[1], Obinna Ekwunife[5], Maureen McGowan[6,7], Margaret Mwangi[3], Agata Muthoni[3], Dorcas Kiboi[3], Sarah Njoroge[3], Fei Gao[8], Jared M. Baeten[2], Kenneth Ngure[2,9]

**1** Public Health Sciences Division, Fred Hutchinson Cancer Center, Seattle, Washington, United States of America, **2** Department of Global Health, University of Washington, Seattle, Washington, United States of America, **3** Center for Clinical Research, Kenya Medical Research Institute, Nairobi, Kenya, **4** Department of Epidemiology, University of Washington, Seattle, Washington, United States of America, **5** Division of Population Health, Department of Medicine, Jacobs School of Medicine and Biomedical Sciences, University at Buffalo, Getzville, New York, United States of America, **6** Heidelberg Institute of Global Health, Heidelberg University, Heidelberg, Germany, **7** South African Centre for Epidemiological Modelling and Analysis, Centre for Epidemic Response and Innovation, School for Data Science and Computational Thinking, Stellenbosch University, Stellenbosch, South Africa, **8** Vaccine and Infectious Diseases Division, Fred Hutchinson Cancer Center, Seattle, Washington, United States of America, **9** School of Public Health, Jomo Kenyatta University of Agriculture and Technology, Nairobi, Kenya

* kortblad@fredhutch.org

## Abstract

### Background

Adolescent girls and young women (AGYW) in Africa experience high HIV acquisition risk and low engagement in prevention services. Knowledge of HIV–negative status paired with peer support might motivate AGYW—who are highly socially connected—to initiate HIV prevention services, including pre-exposure prophylaxis (PrEP).

### Methods and findings

We conducted a randomized controlled trial (ClinicalTrials.gov: NCT04982250) of AGYW peer networks in Central Kenya. Index peers aged 16–24 years who had used oral PrEP in the past 12 months were randomized 1:1 to: (1) *enhanced peer referral:* group training on PrEP referral strategies and delivery of HIV self-testing (HIVST) kits (*n* = 8 kits, 2 kits/peer); or (2) *standard peer referral:* informal PrEP referral strategies. Index peers were encouraged to refer four peers who could benefit from PrEP. Outcomes for referred peers—PrEP initiation (primary), PrEP continuation (i.e., month one refills), and HIV testing (any form following referral)—were reported by index peers three months later. Implementation outcomes and costs were also assessed. Risk differences (RDs) were estimated using generalized linear

**Data availability statement:** All data presented in this paper are freely and publicly available. The deidentified study data, a supporting data dictionary, and the statistical code use for the analyses in this paper can all be found at: https://github.com/Peer-PrEP-trial-Hutch/Peer-PrEP-main-manuscript.

**Funding:** This work was supported by the National Institute of Mental Health (R00 MH121166 to author KFO; https://www.nimh. nih.gov/). The funders had no role in study design, data collection and analysis, decision to publish, or preparation of the manuscript.

**Competing interests:** We have read the journal's policy and the authors of this manuscript have the following competing interests: JMB is an employee of Gilead Sciences, outside the present work, and has no financial stake in the results of this study; KFO is an Academic Editor on PLOS Medicine's editorial board. None of the other authors declare any conflicts of interest.

**Abbreviations:** AGYW, adolescent girls and young women; CIs, confidence intervals; HIVST, HIV self-testing; ICC, intra-cluster correlation coefficient; IQR, interquartile range; IRB, Institutional Review Board; KES, Kenyan Shillings; MD, median difference; MOH, Ministry of Health; PrEP, pre-exposure prophylaxis; RA, research assistant; RDs, risk differences; TFA, Theoretical Framework of Acceptability.

mixed-effects regression models with study group fixed effects and index peer random effects.

From May 3, 2023 to February 16, 2024, 316 index peers were screened and 82 enrolled/randomized (median age 22 years, IQR 20−23): 40 to the enhanced group and 42 to the standard. No index peers were lost to follow-up. Index peers reported outcomes for 241 referred peers (median age 22 years, IQR 21−24): 137 in the enhanced group and 104 in the standard. At follow-up, there were no significant differences in PrEP initiation between the enhanced group (30%, 41/137) and the standard (41%, 41/104; RD −6%, 95% CI [−26%, 11%], $p = 0.51$). Enhanced peer referral was not associated with PrEP continuation (RD 1%, 95% CI [−10%, 13%], $p = 0.82$) but was associated with increased recent HIV testing (RD 39%, 95% CI [24%, 54%], $p < 0.001$). Three referred peers, all in the enhanced group, tested HIV–positive; two social harms (verbal abuse among index peers, one in each group) were reported. Index and referred peers in both study groups found enhanced peer referral acceptable and feasible, but it cost almost eight times more per peer referred to PrEP than standard referral ($23 versus $3 USD). Relying on index peers to report outcomes for referred peers was a study limitation.

## Conclusions

Enhanced peer PrEP referral with training and HIVST delivery did not increase PrEP initiation among Kenyan AGYW but was associated with increased HIV testing, suggesting opportunities to combine peer-delivered HIVST with additional implementation strategies to improve AGYW's PrEP engagement.

---

### Author summary

#### Why was this study done?

- Many adolescent girls and young women (AGYW) throughout Africa have high risk of HIV acquisition and low HIV pre-exposure prophylaxis (PrEP) use.

- AGYWs have close peer networks and prior studies have found that most AGYW who initiate PrEP do so following informal peer referral.

- To increase PrEP initiation among AGYW likely to benefit, we wanted to test the effect of enhancing peer referral with formal referral training and HIV self-testing (HIVST) delivery in Kenya.

#### What did the researchers do and find?

- We conducted a two-arm randomized controlled trial of peer networks in which AGYW PrEP users (i.e., index peers) were encouraged to refer four peers to PrEP (i.e., referred peers) and randomized to either standard or enhanced peer referral.

- Three months following referral, enhanced peer referral did not increase PrEP initiation among referred peers compared to standard peer referral and was not associated with PrEP continuation (i.e., month one refills). However, the intervention was associated with increased recent HIV testing (any form following referral).

- Although enhanced peer referral was delivered with high fidelity and perceived as acceptable, feasible, and appropriate, it was almost eight times the cost of standard referral per peer referred to PrEP.

## What do the findings mean?

- Delivering HIVST as part of enhanced peer PrEP referral did not increase PrEP initiation among AGYW who could benefit in Kenya.

- Peer-delivered HIVST, however, was associated with increased HIV testing uptake, presenting opportunities to layer other strategies—such as remote nurse support, chatbots, peer-delivered PrEP—at the point of testing to increase PrEP awareness and access among AGYW and potentially improve PrEP uptake.

- A study limitation was relying on index peers to report outcomes for referred peers.

## Introduction

Adolescent girls and young women (AGYW, 16–24 years) in many African settings have high risk of HIV acquisition [1,2] and low engagement in highly effective oral HIV pre-exposure prophylaxis (PrEP) services [3–5]. In Kenya, roughly a third of total new HIV infections are among AGYW who account for only ~10% of the population [6], and few AGYW (~5%–15%) at public family planning clinics initiate PrEP when offered [5,7]. Barriers to PrEP uptake among AGYW in Kenya and similar settings are multi-faceted and include individual-level barriers—such as lack of PrEP knowledge, uncertainty about PrEP efficacy and safety, and low HIV risk perception—and community-level barriers—such as limited PrEP access and stigma associated with PrEP use [5,8,9]. To address these barriers and reach more AGYW with PrEP, innovative PrEP referral and delivery strategies catered to this population are needed.

The opinion of peers often influences the health behaviors and preferences of populations with strong social networks, including AGYW. In Africa, most AGYW who initiate PrEP do so because of informal peer referral; in the HPTN 082 open-label PrEP study, 62% of 400 AGYW who initiated PrEP in South Africa and Zimbabwe did so due to peer encouragement [10]. Peer-delivered interventions have successfully increased engagement in HIV prevention services among other populations with strong social networks, including men who have sex with men [11] and female sex workers [12,13], yet remain underexplored among AGYW. HIV prevention interventions that have engaged AGYWs in Africa to date have primarily utilized "PrEP champions" [14], who engage in outreach activities to large peer networks, inclusive of individuals with whom they may or may not have personal connections. Peer delivery of HIV prevention interventions within small, established AGYW peer networks may enable more targeted outreach to individuals known to engage in behaviors associated with HIV risk acquisition who are most likely to benefit.

Strategies that could support peer-delivered HIV prevention interventions within small AGYW peer networks include HIV self-testing (HIVST) and index peer education. HIVST kits have been demonstrated to increase HIV testing uptake and frequency in diverse populations and settings but have been underutilized to support PrEP initiation [15–17]—which could likely benefit many individuals who learn they are HIV–negative following self-testing. To help facilitate linkage to PrEP following HIVST, trusted peers who have used PrEP and received training on PrEP use, safety, and referral could provide counseling at the point of HIVST delivery that addresses known AGYW PrEP initiation barriers, such as PrEP stigma, knowledge, and access.

In collaboration with Kenyan stakeholders—including those from government, professional, and HIV implementing organizations, as well as AGYW—we developed a peer-delivered HIV prevention intervention for small AGYW peer networks that included formal index peer training on PrEP and PrEP referral strategies and delivery of HIVST kits [18]. This intervention was informed by formative qualitative research—which demonstrated high perceived intervention acceptability among Kenyan AGYW and stakeholders [18,19], and pilot testing—which demonstrated feasibility [20]. To test the intervention's effectiveness on PrEP initiation compared to standard informal peer PrEP referral and generate evidence on its implementation that could inform scale-up, we conducted a hybrid randomized controlled trial of AGYW peer networks.

## Methods

This manuscript reports all components of the CONSORT 2025 checklist, S1 CONSORT Checklist.

### Study design and participants

The "Peer PrEP cRCT" was a hybrid type I effectiveness-implementation parallel cluster-level randomized superiority trial of AGYW peer networks (ClinicalTrials.gov: NCT04982250, registered: July 29, 2021) [21,22]. Details of the trial protocol and statistical analysis plan are described elsewhere: S1 and S2 Protocol files and S1 Statistical Analysis Plan. We conducted the trial in three counties in Central Kenya (Kiambu, Nairobi, and Murang'a) where the population-level HIV prevalence is ~2%–4%, slightly lower than the national average of ~4% [23–25].

The AGYW peer networks (i.e., clusters) in this trial consisted of one index peer and up to four referred peers. Eligible index and referred peers were females aged ≥16–24 years, with those <18 years meeting the criteria for emancipated minors (married, mothers, pregnant, or household heads [26,27]). Eligible index peers additionally had used PrEP in the past 12 months, could identify four peers who could benefit from PrEP, were literate, and were not enrolled in another HIV-related study or program. Eligible referred peers additionally had been referred to PrEP by an index peer.

Our study team recruited index peers, who recruited referred peers. To recruit index peers, we reviewed the records of 73 public clinics and four research programs delivering PrEP in the region and called AGYW who met our eligibility criteria. Enrolled index peers were instructed to recruit female peers aged 16–24 years who were PrEP-naïve or had not used PrEP in the past three months, and had not been recruited by another index peer. At the point of PrEP referral, index peers were to encourage referred peers to call study staff to share their contact information so they could be reached for follow-up; if referred peers called staff, both the index and referred peer received 50 Kenyan Shillings (KES; ~$0.30 Dollars [USD]) in phone credits. The Kenya study team carefully monitored the contact information of referred peers to prevent the same individual from being enrolled by more than one index peer. Referred peers could decline study enrollment when later contacted and were ineligible to participate as index peers.

Our trial protocol was approved by the Scientific Ethics Review Unit at the Kenya Medical Research Institute (#4349) and the Institutional Review Board (IRB) at the Fred Hutchinson Cancer Center (#10773). All enrolled participants completed informed consent—either written or verbal (select referred peers only)—administered by Kenyan research assistants (RAs) and received 300 KES (~$1.90 USD) upon completion of each survey. Besides transportation reimbursement for an in-person training (200–700 KES [~$1.30–$4.50 USD] depending on distance, select index peers) and an incentive if referred peers called the study team, index peers received no additional compensation. The trial was overseen by a Data Safety Monitoring Board that met roughly every six months, four times total, over the trial duration.

### Randomization and masking

In this trial, index peers were randomized and outcomes assessed among referred peers—representing clusters of small AGYW peer networks. Index peers were randomized 1:1 to either (1) *enhanced peer referral*, which included an in-person group training on PrEP referral strategies and HIVST delivery (*n* = 8 kits, 2 kits/peer), or (2) *standard peer referral,* which included informal word-of-mouth PrEP referral strategies. Randomization took place at index peers' enrollment visits at the

Partners in Health and Research Development Clinic, a research site in Kiambu County. Index peers' study group assignments were revealed when they opened sealed, opaque envelopes in the presence of research staff. The trial randomization list was prepared using consistent block sizes of four by an experienced data analyst (author CC) who had no direct participant contact. The trial was unblinded due to the nature of the intervention.

## Procedures

Following enrollment, index peers were encouraged to refer four peers they thought might benefit from PrEP using the strategies corresponding to their study group assignment. In both groups, index peers were encouraged to direct referred peers to nearby clinics for PrEP or HIV treatment, if needed. At these clinics, HIV services were delivered without research team support following national guidelines [28]; for PrEP, a one-month drug supply was dispensed at initiation and three-month supply at follow-up. We followed up with all index peers and known referred peers roughly three months later to assess study outcomes.

*Enhanced peer referral:* Index peers assigned to this study group were scheduled to complete a one-time group training on PrEP referral strategies. Research staff experienced in PrEP implementation and AGYW engagement led these half-day trainings of 4–8 index peers on a rolling basis based on the pace of enrollment. Group trainings were selected to save costs and increase index peer engagement. These trainings covered: PrEP use and safety, HIVST use and interpretation, and strategies for engaging peers in conversations around HIV prevention and referring them to HIV services. At the training, the importance of maintaining peers' confidentiality, avoiding stigmatizing language, and addressing HIV myths was emphasized; index peers were also encouraged to disclose their PrEP use to peers, assist peers with HIVST, and escort peers to nearby clinics if they felt comfortable and their peer was interested.

Upon training completion, index peers received a certificate and various tools to support peer PrEP referral. These tools included: (1) AGYW-friendly informational brochures ($n = 4$), with key PrEP and HIVST use/safety facts and a study contact number for questions; (2) HIVST kits ($n = 8$), two of which were to be delivered to each referred peer for personal use or partner testing; and (3) PrEP referral cards ($n = 4$), which were modeled after Kenya Ministry of Health (MOH) referral cards and had information on nearby clinics delivering free HIV services. The blood-based HIVST kits distributed were either the SURE CHECK HIV 1/2 Assay (ChemBio Diagnostics, USA) or Mylan HIV Self Test (Mylan Pharmaceuticals Private Limited, India), based on availability. Additionally, index peers were invited to participate in an optional WhatsApp group, monitored by research staff, which they could contact to address questions or concerns. Beyond the WhatsApp group support, index peers received no additional mentoring or refresher trainings.

*Standard peer referral:* Index peers assigned to this study group did not receive formal training on PrEP referral strategies, informational brochures on PrEP and HIVST, or HIVST kits. At enrollment, they were simply encouraged to discuss PrEP and refer peers to PrEP using the usual approach they would for other sexual health interventions, such as contraception, and were provided the same MOH-style PrEP referral cards ($n = 4$) as the intervention group, reflecting standard peer referral strategies available for AGYW at most public healthcare facilities in Kenya.

*Study visits and assessments:* Over the trial duration, index peers completed two study visits and referred peers one. Index peers completed study visits at enrollment and either three months from their group training (enhanced group) or enrollment (standard group). Referred peers for whom we had contact information completed one study visit—which included enrollment—three months after we received their contact information (ideally at the point of referral). This follow-up period allowed referred peers sufficient time to visit a healthcare facility for PrEP initiation and refills one month later, if interested. At study visits, trained RAs conducted electronic surveys (CommCare, Dimagi, USA) with participants either in-person or over the phone that assessed their demographics (at enrollment), behaviors associated with HIV aquisition risk, PrEP use, and HIV testing history. Additionally, at follow-up, index peers were asked to report characteristics about their referred peers and information on if and how they linked to HIV services. Participants were contacted up to four times before considered lost to follow-up.

For the first five months of trial implementation, we only had the contact information of referred peers who called us at the point of referral. However, after observing low referred peer enrollment, we obtained IRB approval to increase the phone credit compensation to index and referred peers for calling study staff at the point of referral from 50 to 150 KES (~$1.00 USD) and to request the contact information of referred peers from index peers at their follow-up visit.

Roughly eight months into trial implementation, research staff completed time-and-motion observations that assessed the time it took to train index peers in the enhanced group and explain the referral process to index peers in the standard group.

## Outcomes

We report outcomes at follow-up for three study populations: (a) referred peers reported by index peers (in index peer follow-up surveys); (b) referred peers self-reported (in referred peer follow-up surveys); and (c) index peer self-reported (in index peer follow-up surveys). Our prespecified primary outcome was PrEP initiation among referred peers reported by index peers (population a), defined as visiting a clinic or other location and being dispensed PrEP. Prespecified secondary outcomes included recent HIV testing (any form following peer referral) and PrEP continuation (refilling one month following initiation) among referred peers reported by index peers (population a), PrEP adherence (past month) among referred peers (population b), the number of peers referred by each index peer (population c), and PrEP continuation (refilling following enrollment) among index peers (population c). To assess PrEP adherence in the past month, we used a validated three-item 100-point scale [29], with higher scores indicating better adherence and zero indicating no PrEP use. HIV testing, PrEP initiation, and PrEP continuation among referred peers (population b) were prespecified sensitivity analyses that we post-hoc changed to secondary outcomes. Partner HIV testing following referral among referred peers (population b) was added as post-hoc outcome to help understand our null primary finding.

In this study, we also measured several process and implementation outcomes. Process outcomes included what steps participants engaged in (e.g., HIVST use, linkage to care) and support they received following referral. Implementation outcomes included index and referred peers' perceptions of the intervention's acceptability, feasibility, and appropriateness, which we measured using established frameworks [30–32] or scales [32] with five-point Likert scales that assessed level of agreement with various statements. We also measured fidelity by assessing if participants received the materials appropriate for their study group assignment. Additionally, we calculated the cost (in USD) per peer that was referred to and initiated PrEP.

Over the trial duration, we monitored for social harms related to study participation (e.g., verbal, physical, emotional abuse) and serious adverse events (e.g., hospitalization, death). RAs screened for these at study visits and participants could report these over the trial duration by contacting study staff by phone or the WhatsApp group (enhanced index peers only).

## Statistical analysis

The sample size for this trial was based on the primary PrEP initiation outcome and informed by findings from a pilot study of the enhanced peer referral model conducted in the same study setting (in which PrEP initiation among referred peers was 78%, 43/55) [20]. With 80 clusters (40 per study group) of one index peer and three referred peers (75% of those recommended), we had 80% power to detect a 17% difference in PrEP initiation between the standard (63%) and enhanced (80%) groups, assuming an intra-cluster correlation coefficient (ICC) of 0.05—typical for cluster-randomized controlled trials [33]—and an alpha level of 0.05.

We used an intention-to-treat approach for all analyses. In our analyses, "don't know" responses were recoded as failures and only participants with complete data were included (i.e., complete-case analysis). Risk differences (RD) with 95% confidence intervals (CIs) were estimated for all binary outcomes using generalized linear regression models with

Gaussian distributions, identity link functions, study group fixed effects, and index peer random effects (to adjust for clustering). For continuous outcomes, differences in medians were estimated using linear quantile mixed methods with similar specifications.

Prespecified sensitivity analyses included estimating effects: (1) among all potential referred peers (i.e., four per index peer, with missing data considered as failures); (2) among only those who initiated PrEP (PrEP continuation only); (3) with the assumption that "don't know" responses equal success; (4) using multiple imputation to predict "don't know" responses (Markov Chain Monte Carlo models under the assumption of missingness at random [34]); and (5) using generalized estimating equations. All sensitivity analyses were assessed among referred peers reported by index peers (population a). As a post hoc sensitivity analysis, we estimated effects among referred peers (population b) after removing those who reported using PrEP at the point of referral.

As a post hoc exploratory analysis, we conducted bivariable regression models that controlled for study group and adjusted for clustering at the index peer level to understand features of index peers associated with PrEP initiation among referred peers (population a). Features associated with the outcome at the alpha = 0.05 level were considered statistically significant.

To estimate process and implementation outcomes, we used descriptive statistics by study group. We reported the percentage of participants that agreed or strongly agreed with different statements assessing our acceptability, feasibility, and appropriateness outcomes and considered an outcome achieved if ≥80% of participants agreed to most statements (or ≤20% agreed to reverse-coded statements). To calculate costs, we used activity-based micro-costing methods informed by the time-and-motion observations and study records [35], see S1 Table for details. We estimated costs for an MOH scenario that excluded research activities and assumed implementation activities, like training index peers, would be performed by government staff.

We used R version 4.4.1 (R Core Team, 2022) for all analyses.

## Results

From May 3, 2023 to February 16, 2024, we screened 316 index peers and enrolled and randomized 82: 40 to the enhanced group and 42 to the standard, Fig 1. Primary reasons for index peer exclusion were that they could not identify four peers who could benefit from PrEP (41%, 96/234) or were enrolled in another HIV study/program (41%, 95/234). The median age of enrolled index peers was 22 years (interquartile range [IQR] 21–24), roughly a third were married (38%, 31/82), and a fifth in school (20%, 16/82), Table 1.

All index peers were reached for follow-up from September 11, 2023 to June 5, 2024. The median time from enrollment/training to follow-up was 105 days (IQR 91–126) in the standard group and 132 days (IQR 104–167) in the enhanced. At follow-up, index peers reported outcomes for 241 referred peers: 137 in the enhanced group and 104 in the standard. Among these referred peers, the median age was 22 years (IQR 20–23) and roughly half (54%, 130/241) were perceived to have at least one casual sexual partner.

Over the trial duration, we obtained contact information for 60% (145/241) of the referred peers, 116 (48%) of whom we reached for follow-up and 83 (34%) of whom we enrolled from November 21, 2023 to August 27, 2024. We encountered several challenges reaching referred peers for follow-up, including few calling us at the point of referral (prior to an IRB modification that enabled us to contact them directly) and few answering their phone when we attempted to reach them. Common reasons for exclusion among those reached were age ineligibility (52%, 17/33) and disinterest in study participation (36%, 12/33). All enrolled referred peers were unique individuals; 55 in the enhanced group and 28 in the standard. Among referred peers enrolled, the median age was 22 years (IQR 20–23), roughly a fifth were married (18%, 15/83), and almost half were in school (44%, 36/83). At enrollment, almost all referred peers (95%, 79/83) reported condomless sex in the past six months with a partner whose HIV status was unknown or who was living with HIV.

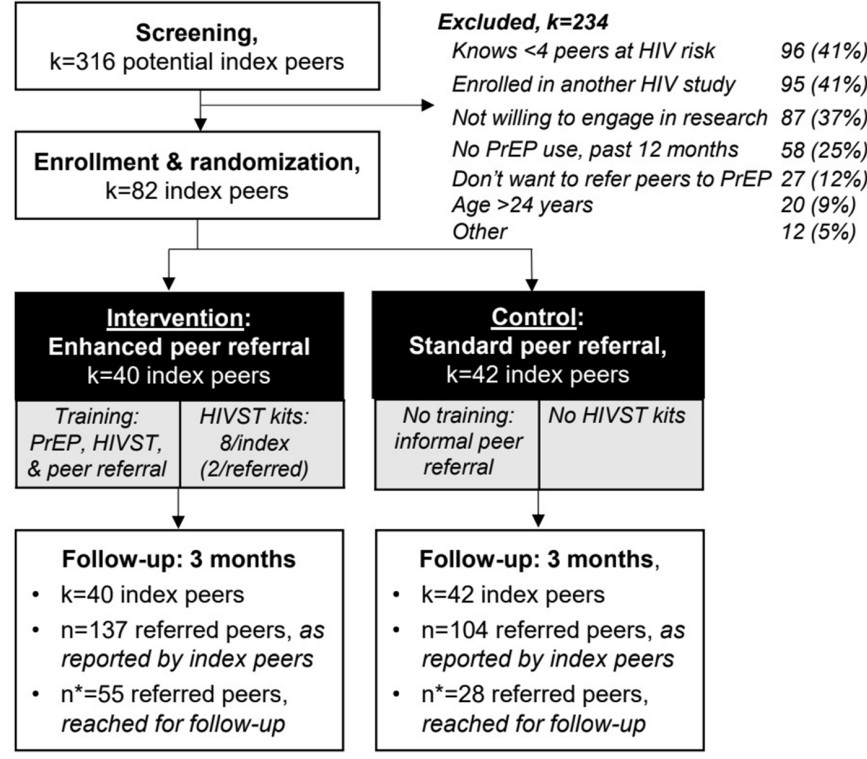

**Fig 1. CONSORT diagram of study participants: index peers (k) and referred peers (n and n*).**

At follow-up, roughly three months from index peers' enrollment (standard group) or training (enhanced group), there were no statistically significant differences in PrEP initiation among referred peers; with index peers reporting that 30% (41/137) of referred peers in the enhanced group and 41% (41/104) in the standard group initiated PrEP (RD −6%, 95% CI [−26%, 11%], p = 0.51; estimated ICC 0.66), Fig 2. Additionally, enhanced peer referral was not associated with PrEP continuation among referred peers at follow-up (enhanced: 13%, 17/137; standard: 11%, 11/104; RD 1%, 95% CI [−10%, 13%], p = 0.82) but was associated with increased recent HIV testing (enhanced: 95%, 125/137; standard: 54%, 56/104; RD 39%, 95% CI [24%, 54%], p < 0.001). According to index peers, three referred peers tested HIV–positive: all in the enhanced group. These findings remained largely consistent in our sensitivity analyses, S2 Table. In our exploratory analysis, PrEP use among index peers at the point of peer referral was associated with increased PrEP initiation among referred peers (RD 22%, 95% CI [3%, 42%], p = 0.03), S3 Table; no other index peer features were associated with this outcome.

While index peers in the enhanced group reported referring more peers to PrEP (median: 4 referred peers, IQR 3–4) than those in the standard group (median: 3, IQR 3–4), the intervention was not associated with more PrEP referrals (RD 0.5, 95% CI [−0.2, 1.1], p = 0.09), Table 2. When we evaluated PrEP initiation, PrEP continuation, and HIV testing among referred peers reached for follow up, our findings were consistent: there were no significant differences in PrEP initiation

PLOS Medicine

**Table 1. Demographics and sexual behaviors of index and referred peers, as reported by the respective participant groups.**

| Characteristics | Index peers | | Referred peers | | | |
| --- | --- | --- | --- | --- | --- | --- |
| | Self-reported | | Reported by index peers | | Self-reported | |
| | Enhanced referral, k=40 | Standard referral, k=42 | Enhanced referral, n=137 | Standard referral, n=104 | Enhanced referral, n*=55 | Standard referral, n*=28 |
| *Demographics* | | | | | | |
| Age; med (IQR) | 22 [20; 23] | 22 [21; 24] | 21 [20; 23] | 22 [20; 23] | 22 [20; 23] | 22 [20; 23] |
| Married | 14 (34%) | 17 (41%) | | | 10 (18%) | 5 (18%) |
| Relationship status: | | | | | | |
| One primary partner | 27 (66%) | 28 (67%) | 43 (32%) | 31 (30%) | 34 (62%) | 11 (39%) |
| Any casual partners | 9 (23%) | 9 (21%) | 72 (53%) | 58 (55%) | 14 (26%) | 9 (32%) |
| Single, no partners | 4 (10%) | 5 (12%) | 22 (16%) | 11 (11%) | 7 (13%) | 8 (29%) |
| Other | 0 | 0 | 0 | 4 (4%) | 0 | 0 |
| Years of school; med (IQR) | 12 [12; 14] | 12 [10; 13] | | | 12 [11; 13] | 13 [12; 14] |
| Currently in school | 8 (20%) | 8 (19%) | | | 22 (40%) | 14 (50%) |
| Monthly income; med (IQR) | 2500 [0; 15000] | 999 [0; 7375] | | | 0 [0; 6000] | 0 [0; 6250] |
| Currently enrolled in another HIV study | 0 | 0 | | | 5 (9%) | 2 (7%) |
| *Sexual and reproductive behaviors* | | | | | | |
| Age of first sex; med (IQR) | 18 [16; 19] | 18 [17; 20] | | | 18 [17; 19] | 18 [17; 20] |
| No. of sex partners, past 3 mos; med (range) | 1 [0; 5] | 1 [0; 2] | | | 0 [0; 2.5] | 0 [0; 1.3] |
| New sexual partner, past 3 mos | 11 (28%) | 10 (24%) | | | 3 (6%) | 1 (4%) |
| Ever being pregnant | 29 (73%) | 32 (76%) | | | 25 (46%) | 10 (36%) |
| Using a LARC | 20 (50%) | 21 (50%) | | | 14 (26%) | 7 (25%) |
| EC use, past 6 mos | 31 (78%) | 34 (81%) | | | 8 (15%) | 5 (19%) |
| EC use ≥2×, past 6 mos | 2 (5%) | 2 (5%) | | | 3 (5×) | 3 (11%) |
| *Behaviors associated with HIV, past 6 months* | | | | | | |
| Condomless sex with partner of unknown or positive HIV status | 25 (63%) | 32 (76%) | | | 52 (95%) | 27 (96%) |
| STI diagnosis or treatment | 6 (15%) | 7 (17%) | | | 5 (9%) | 7 (25%) |
| PEP uses ≥2 times | 0 (0%) | 2 (5%) | | | 0 | 0 |
| Engaged in sex with drugs or alcohol | 12 (30%) | 8 (19%) | | | 10 (18%) | 8 (29%) |
| Engaged in transactional sex | 6 (15%) | 9 (21%) | 49 (37%) | 48 (53%) | 7 (13%) | 7 (25%) |
| *Prior PrEP use* | | | | | | |
| PrEP use: any use prior to peer referral | 40 (100%) | 42 (100%) | | | 10 (18%) | 9 (32%) |
| PrEP use: at the point of peer referral | 27 (68%) | 31 (74%) | | | 6 (11%) | 6 (24%) |

**Abbreviations:** number referred peers with outcomes reported by index peers (*n*); number referred peers with outcomes reported by referred peers (*n**); number of index peers with outcomes reported by index peers (*k*); interquartile range (IQR); emergency contraception (EC); long-acting injectable contraception (LARC); post-exposure prophylaxis (PEP); pre-exposure prophylaxis (PrEP); risk assessment screening tool (RAST); sexually transmitted infections (STI).

between the study groups (RD −24%, 95% CI [−50%, 1%], $p=0.07$), and enhanced peer referral was not associated with PrEP continuation (RD −9%, 95% CI [−25%, 8%], $p=0.33$) but was associated with increased recent HIV testing (RD 36%, 95% CI [21%, 51%], $p<0.001$). Additionally, enhanced peer referral was not associated with partner HIV testing among referred peers (RD 11%, 95% CI [−17%, 38%], $p=0.46$), PrEP adherence among referred peers (RD −7.5, 95% CI [−28.1, 13.2], $p=0.47$), and PrEP continuation among index peers (RD −19%, 95% CI [−40%, 2%], $p=0.08$). Only one referred peer in the standard group reported a partner testing HIV-positive following referral.

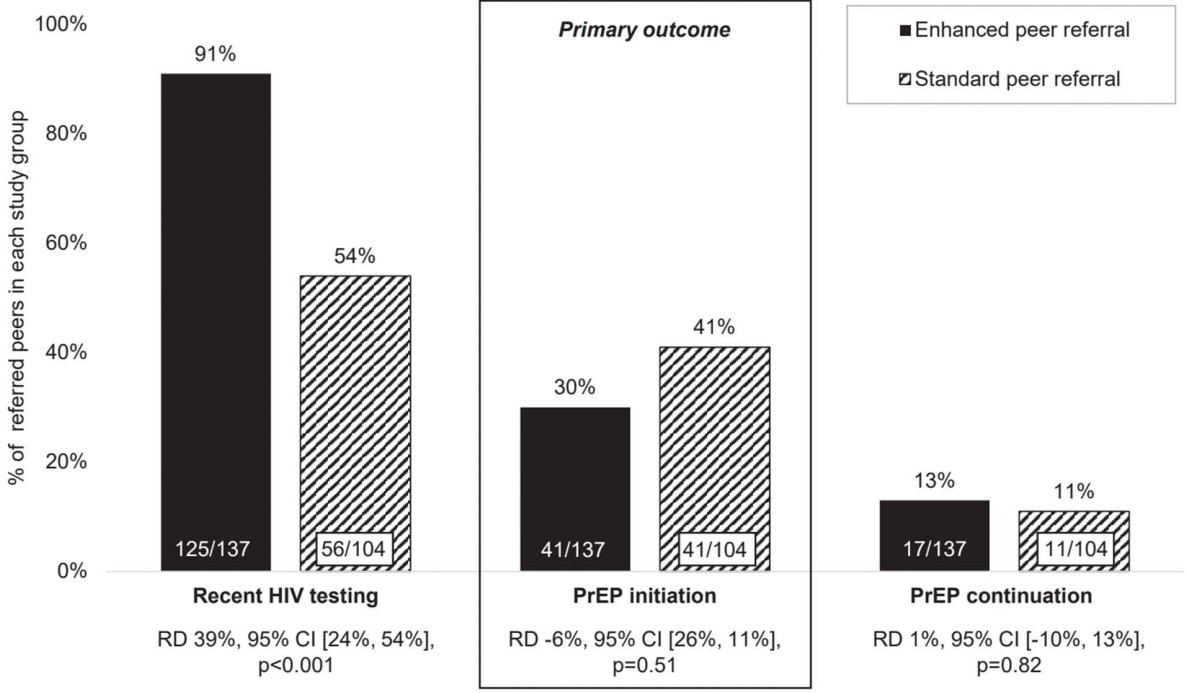

**Abbreviations:** Confidence interval (CI); pre-exposure prophylaxis (PrEP); risk differences (RD).

**Fig 2. The effect/association of enhanced vs. standard peer referral on HIV testing, PrEP initiation, and PrEP continuation among referred three months later, as reported by index peers.**

Our process outcomes suggest that in both study groups engagement in peer PrEP referral was high, Table 3. Index peers reported disclosing their PrEP use to most referred peers (90%, 216/241) and escorted roughly one in five (17%, 40/241) to clinic-based HIV services. Almost all referred peers in the enhanced group (98%, 52/55) reported using an HIVST kit for self-testing and nearly half (42%, 23/55) reported using the other for partner testing. Compared to the enhanced group, more referred peers in the standard group reported visiting a clinic for HIV services (61% versus 31%) and being escorted there by an index peer (21% versus 9%).

Formalized peer PrEP referral supported with HIVST delivery was perceived as highly acceptable, feasible, and appropriate by index and referred peers in both study groups; assigned services in both study groups were also delivered with high fidelity. Most peers reported liking the intervention (>88%) and believed it would help AGYW remain HIV-negative (>90%). Few thought it was hard to deliver (<11%) or felt it interfered with their other priorities (<2%). Most thought the intervention was possible to implement (>90%) and fit the HIV prevention needs of young women in Kenya (>95%). Additionally, most referred peers reported receiving all interventional materials from index peers (84%, 70/83); only one referred peer in the enhanced group reported not receiving two HIVST kits.

Time-and-motion observations revealed that it took ~4 hours to train index peers on PrEP referral in the enhanced group and ~8 min to encourage index peers to refer peers to PrEP in the standard group. In our economic analysis, the cost per peer referred to PrEP was higher in the enhanced group ($23 USD) compared to the standard group ($3 USD), as was the cost per peer initiated on PrEP in the enhanced group ($77 USD) compared to the standard group ($7 USD). Higher costs in the enhanced group were largely driven by the cost of HIVST kits (>36% of total costs) and other recurrent supplies (35%), S1 Table.

**Table 2. The association of enhanced vs. standard peer referral on secondary and exploratory outcomes among referred and index peers three months later.**

| | Enhanced peer referral | Standard peer referral | Risk and median differences [95% CI] | *p*-value |
|---|---|---|---|---|
| **Among referred peers, *reported by index peers*** | ***n* = 137** | ***n* = 104** | | |
| Number of referred peers, MD[1,2] | 4.0 [3.0; 4.0] | 3.0 [3.0; 4.0] | 0.5 [−0.2, 1.1] | 0.09 |
| **Among referred peers, *self-reported*** | ***n\* = 55*** | ***n\* = 28*** | | |
| Recent HIV testing, RD[3] | 54 (98%) | 17 (61%) | 36% [21%, 51%] | <0.001 |
| Partner HIV testing, RD[4] | 25 (46%) | 6 (35%) | 11% [−17%, 38%] | 0.46 |
| PrEP initiation, RD[3] | 13 (24%) | 12 (43%) | −24% [−50%, 1%] | 0.07 |
| PrEP continuation, RD[3] | 6 (11%) | 6 (21%) | −9% [−25%, 8%] | 0.33 |
| PrEP adherence score, MD[1,5] | 1.1 [0; 33.3] | 0 [0; 29] | −7.5 [−28.1, 13.2] | 0.47 |
| **Among index peers, *self-reported*** | ***k* = 40** | ***k* = 42** | | |
| PrEP continuation, RD | 19 (48%) | 28 (67%) | −19% [−40%, 2%] | 0.08 |

[1]For MDs, medians and interquartile ranges reported.

[2]A prespecified process vs. secondary outcome we decided to assess as a secondary outcome.

[3]A prespecified sensitivity vs. secondary outcome we decided to assess as a secondary outcome.

[4]A non-prespecified exploratory outcome we added post-hoc to understand our null PrEP initiation (primary) findings.

[5]PrEP adherence assessed using a 100-point scale that is an average of participants' responses to three questions on adherence in the past 30 days: (1) # of missed pills [(30 days-# pills)*3.33], 2) self-reported PrEP adherence (on 5-point Likert scale, 0–4 points*25), and (2) self-reported frequency of PrEP use (on 5-point Likert scale, 0–4 points*25). PrEP adherence score = ($a + b + c$)/3 (Wilson I and colleagues. AIDS and Behavior 2014).

Abbreviations: confidence interval (CI); median difference (MD); number referred peers with outcomes reported by index peers (*n*); number referred peers with outcomes reported by referred peers (*n\**); number of index peers with outcomes reported by index peers (*k*); pre-exposure prophylaxis (PrEP); risk difference (RD).

Two social harms (both verbal abuse among index peers, one in each group) and no serious adverse events were reported over the trial duration; both social harms were related to study participation.

## Discussion

Among Kenyan AGYW at risk of HIV acquisition, enhanced peer PrEP referral supported with index peer training and HIVST delivery did not significantly increase PrEP initiation compared to standard, word-of-mouth peer referral. Additionally, the intervention was not associated with one-month PrEP continuation but was associated with increased recent HIV testing. Peers who delivered and received the intervention perceived it as acceptable and feasible, however, enhanced peer referral was roughly eight times the cost of standard peer referral for each peer referred to PrEP. These findings highlight opportunities for new implementation strategies—such as telehealth visits or direct peer PrEP delivery [36–40]—to be paired with peer-delivered HIVST kits to support linkage from at-home testing to clinic-based PrEP among AGYW who test HIV-negative, the vast majority.

In this intervention, the use of HIVST to promote linkage to prevention services was novel, as most of HIVST interventions to date have primary focused on identifying individuals living with HIV and facilitating linkage to treatment services [41–44]. However, most individuals who complete HIVST at home will test negative and may feel motivated to change their behavior and/or engage in HIV prevention interventions to maintain their HIV–negative status [45,46]. Consistent with other peer-delivered HIVST interventions in populations with strong social networks [14–16], this intervention was associated with increased HIV testing uptake among AGYW. The index peer training on PrEP referral strategies, however, was not enough to translate HIV testing increases into PrEP initiation increases compared to standard, informal peer referral. For AGYW in the enhanced group, barriers to clinic-based PrEP remained; while for AGYW in the standard group, many

**Table 3. Process and implementation outcomes related enhanced and standard peer PrEP referral among referred and index peers at 3-month follow-up.**

| | Enhanced peer referral | Standard peer referral |
|---|---|---|
| **Process outcomes** | | |
| **Among referred peers, *reported by index peers*** | **n = 137** | **n = 104** |
| Days from index peer training/enrollment to peer referral, median (IQR) | 5 [2; 13] | 15 [3; 60] |
| Disclosed PrEP use to referred peer | 123 (90%) | 93 (89%) |
| Escorted referred peer to clinic for HIV services | 22 (16%) | 18 (17%) |
| **Among referred peers, *self-reported*** | **n* = 55** | **n* = 28** |
| Used an HIVST kit for self-testing | 54 (98%) | N/A |
| *Used an HIVST kit for self-testing twice[1]* | 4 (1%) | N/A |
| *Used an HIVST kit for partner testing[1]* | 23 (42%) | N/A |
| *Index peer assisted with HIVST[1]* | 32 (58%) | N/A |
| Visited clinic for HIV services | 17 (31%) | 17 (61%) |
| *Escorted to clinic by index peer* | 5 (9%) | 6 (21%) |
| **Implementation outcomes** | | |
| **Among index peers, *self-reported*** | **k = 40** | **k = 42** |
| **Acceptability[2,3]** | | |
| I liked referring my peers to PrEP services with HIV self-tests to support referral *(TFA: Affective attitude)* | 35 (88%) | 41 (98%) |
| It is hard to refer peers to PrEP and deliver HIVST kits *(TFA: Burden)* | 4 (10%) | 2 (5%) |
| I was able to carry-out all intervention elements successfully *(TFA: Self-Efficacy)* | 37 (92%) | 41 (98%) |
| I was able to solve problems that arose while delivering the intervention *(TFA: Self-Efficacy)* | 35 (88%) | 40 (95%) |
| I don't understand why should refer peers to PrEP and deliver HIVST kits *(TFA: Intervention coherence)* | 1 (2%) | 2 (5%) |
| Referring peers to PrEP with HIVST kits interfered with other priorities *(TFA: Opportunity Costs)* | 0 | 0 |
| Referring peers to PrEP with HIVST kits will help peers remain HIV–negative *(TFA: Perceived effectiveness)* | 36 (90%) | 42 (100%) |
| **Feasibility[3,4]** | | |
| Referring peers to PrEP with HIVST kits is possible to implement | 36 (90%) | 38 (90%) |
| It is doable to refer peers to PrEP and deliver HIVST kits to support referral | 37 (92%) | 39 (93%) |
| **Appropriateness[3,5]** | | |
| Referring peers to PrEP with HIVST kits fits the HIV prevention needs of young women in Kenya | 38 (95%) | 42 (100%) |
| Referring peers to PrEP with HIVST kits is a good match for young women seeking prevention services | 38 (95%) | 41 (98%) |
| **Among referred peers, *self-reported*** | **n* = 55** | **n* = 28** |
| **Acceptability[2,3]** | | |
| I liked being referred to PrEP by peers with HIVST kits to support the referral *(TFA: Affective attitude)* | 55 (100%) | 28 (100%) |
| It was hard being referred to PrEP by peers with HIVST kits to support the referral *(TFA: Burden)* | 6 (11%) | 1 (4%) |
| I am confident the peer who referred me carried-out all intervention elements successfully *(TFA: Self-Efficacy)* | 54 (98%) | 26 (93%) |
| I am confident the peer who referred me could solve problems related to intervention delivery *(TFA: Self-Efficacy)* | 52 (95%) | 25 (89%) |
| I don't understand why a peer referred me PrEP with HIVST kits *(TFA: Intervention coherence)* | 1 (2%) | 0 |
| Being referred PrEP by a peer with HIVST kits interfered with my other priorities *(TFA: Opportunity Costs)* | 0 | 1 (4%) |
| Peer PrEP referral with HIVST delivery will help my peers remain HIV–negative *(TFA: Perceived effectiveness)* | 55 (100%) | 28 (100%) |
| **Feasibility[3,4]** | | |
| Peer PrEP referral with HIVST kit delivery was possible to implement | 54 (98%) | 27 (96%) |
| It was doable for my peers to refer me to PrEP and deliver HIVST kits to support the referral | 55 (100%) | 27 (96%) |

*(Continued)*

**Table 3.** (Continued)

| | Enhanced peer referral | Standard peer referral |
|---|---|---|
| **Appropriateness**[3,5] | | |
| Referring peers to PrEP with HIVST kits fits the HIV prevention needs of young women in Kenya | 38 (95%) | 42 (100%) |
| Referring peers to PrEP with HIVST kits is a good match for young women seeking prevention services | 38 (95%) | 41 (98%) |
| **Fidelity** | | |
| Received all materials[6] from index peers | 46 (84%) | 24 (86%) |
| *Received PrEP/PEP educational brochure* | 46 (84%) | N/A |
| *Received two HIVST kits* | 54 (98%) | N/A |
| *Received referral card to PrEP services* | 47 (85%) | 24 (86%) |
| **Cost**[7] | | |
| Per peer referred to PrEP | $23 | $3 |
| Per peer who initiated PrEP | $77 | $7 |

[1]Process outcomes reported only among referred peers in the enhanced referral group (i.e., intervention group).

[2]Based on the TFA (Sekhon and colleagues [31]).

[3]Each statement was measured using a 5-point Likert agreement scale; we report the proportion that agreed/strongly agreed.

[4]Based on the Feasibility of Intervention Measurement (FIM) (Weiner and colleagues [32]).

[5]Based on the Intervention Appropriateness Measure (IAM) (Weiner and colleagues [32]).

[6]Materials included: educational brochure, two HIVST kits, referral card to PrEP services.

[7]Costs reported for a feasible Ministry of Health scale-up scenario, with salary costs included from government salaries and research-related costs removed.

Abbreviations: HIV self-testing (HIVST); number referred peers with outcomes reported by index peers ($n$); number referred peers with outcomes reported by referred peers ($n*$); number of index peers with outcomes reported by index peers ($k$); pre-exposure prophylaxis (PrEP); Theoretical Framework of Acceptability (TFA).

of the barriers they had to overcome for clinic-based testing were like those for clinic-based PrEP, facilitating same-day PrEP initiation.

Differences in partner HIV testing among referred peers in the two study groups might have contributed to our null PrEP initiation findings. In the enhanced group, nearly half of referred peers reported using their second HIVST kit for partner testing (42%), which may have contributed to higher partner testing uptake following peer referral in the enhanced group (46%) compared to the standard group (35%), though the intervention was not associated with increased partner testing. These findings would be consistent with other studies in Kenya and Malawi that demonstrated delivery of multiple HIVST kits to pregnant women significantly increased male partner HIV testing [47,48]. In this study, most referred peers who engaged in partner HIV testing learned their partner was not living with HIV, which may have demotivated their PrEP interest.

The null findings in this study might also be attributable to differences in the characteristics of peers referred in the enhanced group compared to the standard. Index peers in the enhanced group referred more peers to PrEP than those in the standard group, though this difference was not associated with the intervention. Nonetheless, referred peers in the standard group might have been at higher risk of HIV acquisition than those in the enhanced group. It is possible that the training and HIVST kits provided to index peers in the enhanced group created additional motivation, or pressure, to meet the referral target of four peers, potentially leading them to refer individuals who were less likely to benefit from PrEP. This could have driven down the proportion of referred peers who initiated PrEP. Limited data on referred peer characteristics, however, prevented us from adequately assessing this possibility.

This trial had strengths and weaknesses. Strengths included 100% follow-up completion among index peers; reporting outcomes among referred peers using data from both index peers (primary) and referred peers (secondary); and presenting both effectiveness and implementation outcomes. Weaknesses included relying on index peers to recruit unique and eligible referred peers, risking potential contamination between study groups—which we monitored for and did not observe. Another weakness included relying on index peers to report characteristics and outcomes for referred peers, limiting our ability to describe the population referred and subjecting our outcomes to potential missingness, recall bias, and falsification. Despite the limitations of this approach, having index participants report outcomes for recruited participants is common practice in assessment of peer-delivered interventions [47,48]. While our study findings remained consistent among the referred peers reached for follow-up, we encountered challenges reaching this population, which resulted in high and differential loss-to-follow-up by study group (enhanced: 60%; standard: 73%) that may have biased our secondary outcomes and estimated associations. The relatively short follow-up period (three months) was another study weakness, as this covered only one PrEP refill opportunity and prevented assessment of the intervention's impact on referred peers' participation in HIV prevention services over time. Finally, the observed ICC (0.66) was substantially larger than that anticipated in the study design (0.05), which reduced our statistical power to detect significant differences in PrEP initiation between study groups. This finding has important implications for the design of future studies examining interventions within small AGYW peer networks, as higher ICCs require more clusters or larger sample sizes to maintain adequate power.

As we enter a new era of constrained funding for HIV programming, low-cost implementation strategies that can target intervention delivery to those who could benefit most are needed [49]. Consistent with our pre-trial assumption, standard informal PrEP referral within small peer networks resulted in high PrEP initiation (41%)—much higher than that observed at family planning clinics in Kenya (4%–16% [5,7]), suggesting that peer delivery of MOH PrEP referral cards alone—a light touch intervention—might be effective for engaging AGYW in PrEP [50]. While peer PrEP referral enhanced with training and HIVST delivery did not increase PrEP initiation compared to standard referral, it was associated with increased HIV testing—the first step of the HIV prevention and treatment cascade. Future intervention adaptations should therefore consider strategies to better support linkage to HIV services, including strategies that might enhance the quality of counseling services received at the point of referral—such as telehealth visits [38,51] or artificial intelligence-assisted chatbots [52]—or strategies that bring services directly to where AGYW are testing—such as peer-delivered or courier-delivered PrEP [37,38,53], likely following a telehealth visit. These findings can inform future interventions addressing HIV prevention coverage gaps among AGYW in Kenya and similar settings, helping HIV programs optimize service delivery.

## Supporting information

**S1 CONSORT Checklist.** Hopewell S, Chan AW, Collins GS, Hróbjartsson A, Moher D, Schulz KF, et al. CONSORT 2025 Statement: updated guideline for reporting randomized trials. BMJ. 2025; 388:e081123. https://dx.doi.org/10.1136/bmj-2024-081123 2025 Hopewell and colleagues. This is an Open Access article distributed under the terms of the Creative Commons Attribution License.
(PDF)

**S1 Protocol. Peer PrEP referral + HIV self-test delivery for PrEP initiation among young Kenyan women: Pilot study and randomized implementation trial** *(Final version)***.**
(PDF)

**S2 Protocol. Peer PrEP referral + HIV self-test delivery for PrEP initiation among young Kenyan women: Pilot study and randomized implementation trial** *(Original version).*
(PDF)

**S1 Statistical Analysis Plan. Peer PrEP referral + HIV self-test delivery for PrEP initiation among young Kenyan women: Pilot study and randomized-controlled implementation trials STATISTICAL ANALYSIS PLAN.**
(PDF)

**S1 Table. Costing inputs that informed our unit cost estimates in the Ministry of Health scenario.**
(DOCX)

**S2 Table. Sensitivity analyses for effect size estimates among referred peers reported by index peers (n) or referred peers self-reported (*n\**).**
(DOCX)

**S3 Table. Features of index peers associated with PrEP initiation among referred peers, as reported by index peers (*n*).**
(DOCX)

## Acknowledgments

We would like to acknowledge all the study participants, including the index peers who referred peers to PrEP and the index and referred peers who completed study activities. Additionally, we would like to acknowledge the teams at the Kenya Medical Research Institute and Fred Hutchinson Cancer Center who supported participant recruitment and training, data collection, and grant management, including Kendall Harkey for her administrative support.

## Author contributions

**Conceptualization:** Katrina F. Ortblad, Maureen McGowan, Jared M. Baeten, Kenneth Ngure.

**Data curation:** Carlos Culquichicon, Irene Njeru.

**Formal analysis:** Katrina F. Ortblad, Carlos Culquichicon, Obinna Ekwunife, Fei Gao, Kenneth Ngure.

**Funding acquisition:** Katrina F. Ortblad, Jared M. Baeten.

**Investigation:** Katrina F. Ortblad, Kenneth Ngure.

**Methodology:** Katrina F. Ortblad, Obinna Ekwunife, Fei Gao, Jared M. Baeten, Kenneth Ngure.

**Project administration:** Njeri Wairimu, Rachel C. Malen, Adriana M. Reedy, Margaret Mwangi, Agata Muthoni, Dorcas Kiboi, Sarah Njoroge, Kenneth Ngure.

**Resources:** Katrina F. Ortblad, Kenneth Ngure.

**Supervision:** Katrina F. Ortblad, Rachel C. Malen, Fei Gao, Jared M. Baeten, Kenneth Ngure.

**Validation:** Katrina F. Ortblad, Njeri Wairimu, Carlos Culquichicon, Irene Njeru, Maureen McGowan, Margaret Mwangi.

**Visualization:** Katrina F. Ortblad, Carlos Culquichicon, Adriana M. Reedy.

**Writing – original draft:** Katrina F. Ortblad, Njeri Wairimu, Carlos Culquichicon, Irene Njeru, Rachel C. Malen, Adriana M. Reedy, Obinna Ekwunife, Maureen McGowan, Margaret Mwangi, Agata Muthoni, Dorcas Kiboi, Sarah Njoroge, Fei Gao, Jared M. Baeten, Kenneth Ngure.

**Writing – review & editing:** Katrina F. Ortblad, Njeri Wairimu, Carlos Culquichicon, Irene Njeru, Rachel C. Malen, Adriana M. Reedy, Obinna Ekwunife, Maureen McGowan, Margaret Mwangi, Agata Muthoni, Dorcas Kiboi, Sarah Njoroge, Fei Gao, Jared M. Baeten, Kenneth Ngure.

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
