## [Editor Report · Decision Letter 0]

15 Jul 2025

Dear Dr Ortblad,

Sincere apologies for the delay in getting back to you with a decision and thank you for submitting your manuscript entitled "The effect of enhanced peer PrEP referral with HIV self-testing delivery among young Kenyan women: A cluster-randomized controlled trial" for consideration by PLOS Medicine.

Your manuscript has now been evaluated by the PLOS Medicine editorial staff with relevant expertise and I am writing to let you know that we would like to send your submission out for external peer review.

For clinical studies, please upload a copy of your trial study protocol as a supporting information file. The study protocol should be the version submitted for approval to the institutional review board or ethics committee, should include any amendments to the study protocol, as well as the date of their approval by the institutional review or ethics committee. Please also detail any deviations from the study protocol in the Methods section of your manuscript. The editors will consider the protocol and study conduct prior to a final decision for external review. Please also add the dates of enrollment of the first and last participants to the Methods section, as well as the inclusion and exclusion criteria.

Please re-submit your manuscript within two working days, i.e. by Jul 17 2025 11:59PM.

Kind regards,

Andreia Cunha, PhD

Senior Editor

PLOS Medicine

---

## [Decision Letter · Decision Letter 1]

16 Sep 2025

Dear Dr Ortblad,

Sincere apologies for the delay in getting back to you with a decision, which was due to challenges in securing Reviewers. Many thanks for submitting your manuscript "The effect of enhanced peer PrEP referral with HIV self-testing delivery among young Kenyan women: A cluster-randomized controlled trial" (PMEDICINE-D-25-02363R1) to PLOS Medicine. The paper has been reviewed by subject experts and a statistician; their comments are included below and can also be accessed here: [LINK]

As you will see, the reviewers find your work of considerable interest, but they raise some important points that we would like to see fully resolved on resubmission. After discussing the paper with the editorial team and an academic editor with relevant expertise, I'm pleased to invite you to revise the paper in response to the reviewers' comments. We plan to send the revised paper to some or all of the original reviewers, and we cannot provide any guarantees at this stage regarding publication. Please be advised that we may invite a third independent reviewer to consider a revised manuscript.

We ask that you submit your revision by Oct 16 2025 11:59PM. However, if this deadline is not feasible, please contact me by email, and we can discuss a suitable alternative.

Don't hesitate to contact me directly with any questions (acunha@plos.org).

Best regards,

Andreia

Andreia Cunha, PhD

Senior editor

PLOS Medicine

acunha@plos.org

Comments from the academic editor:

- Consider a more descriptive study design for title. Although I understand the social network clusters are randomized, it does seem like this is somewhat more nuanced design with individual randomization of index peers and then assessing outcomes amongst their network.

- Some of the discussion of innovation of intervention may be better suited for discussion (after it is clear what the intervention are).

- Some reoorganization/clarity in the methods would be helpful. It took awhile to get oriented to the design and intervention (e.g., index peers randomized, outcome is among referred peers; intervention is enhanced training and HIVST, etc.) For example, initially was not clear how far out in the social network was going to be studied or whether referred peers would also be part of intervention and HIVST distribution, etc.

- Were there any positive HIV tests?

- Were there are feature of index peers or their referrals associated with better outcomes?

Comments from the reviewers:

Reviewer #1: Statistical review

This paper reports a cluster randomised trial comparing approaches to utilising peers to encourage people to engage with pre-exposure preventative treatment for HIV in Kenya. Overall I thought the trial was well reported. I have some relatively minor comments on the statistical aspects below.

1. Abstract: I would recommend that (for at least the primary outcome), that a p-value is given in addition to the confidence interval, given the trial was prospectively powered to test that outcome's hypothesis. Personally I'd also provide p-values for secondary endpoints (especially as one of them has a significance claim provided) but the authors may feel that this is less appropriate.

2. Page 7: "The trial protocol and statistical analysis plan can be found on the trial registration website" - I am not sure if these are separate documents but I could not see the SAP provided on ct.gov (there is a protocol and informed consent form listed at the bottom of the page). Is there a SAP that could be provided as a supplementary file?

3. Page 11: I did not find it clear what the k, n, n* notation meant (until later in the results). Perhaps it could be briefly clarified that this the size of the population being referred to?

4. Page 11: there are several endpoints mentioned on the clinicaltrials.gov page which I didn't see reported here (e.g. social support, PrEP knowledge). I would recommend that the paper mentions all outcomes in the clinicaltrials.gov page, even if they are not reported here.

5. Page 12: "identify links' should be 'identity link function'.

6. Page 13: I would have welcomed more mention of the reasons for the big difference between n and n* - is it definitely confirmed that referred peers as reported by the index peers are genuine? Where the referred peer was followed-up with, did information reported by the index peer generally agree with the information reported by the referred peer?

7. Table 2: as per my comment on the abstract, I feel p-values are appropriate to add to results, especially as the statistical section refers to significance level.

8. Results: I would recommend reporting the estimated ICC, as this could be useful for future studies.

9. Page 19 - the last paragraph implies that the rate of PrEP initiation was higher than expected. Unless I misunderstood, it was considerably lower than what was anticipated in the power calculation, so I didn't follow this statement.

James Wason

Reviewer #2: Summary

The effect of enhanced peer PrEP referral with HIV self-testing delivery among young Kenyan women: A cluster-randomized controlled trial

This cluster-randomized controlled trial assesses the effects of enhanced peer referral compared to standard peer referral on PrEP initiation among adolescent girls and young women in Kenya. This study aims to address an important public health challenge associated with low PrEP uptake and continuation among a group disproportionately impacted by HIV. I have some comments as follows.

Abstract

1. Line 71- specify timeframe for the PrEP continuation (month 1 refill)

Methods

1. Line 118 and throughout the text- Specify type of hybrid design (type 1?)

2. Lines 122-123 "We conducted this trial in three counties in Central Kenya (Kiambu, Nairobi, and Murang'a) where the 123 population-level HIV prevalence is ~2% [22]." Contrast this data with the national estimate to provide further background information to the readers.

3. Specify how many facilities were included to review the records of public clinics.

4. Lines 169-170- "Research staff experienced in PrEP implementation and AGYW engagement led 170 these one-day trainings of 4-8 index peers…". Can you specify if this one-day training was the only training received? Any additional mentoring or refresher training conducted during implementation?

5. How long was the training delivered to the peers assigned to the control group?

6. Lines 213- for HIV testing, I am assuming this refers to HIV testing at the clinic and not HIVST. Please specify in the text.

7. Line 237- provide justification or citation for the intra-cluster correlation value.

8. For referred peers, it would be helpful if the authors specify how they confirm the identity of the referred peers to prevent being referred twice (and counted twice) by different index peers.

9. Include brief information about the sample size calculation.

Discussion

1. Lines 331- specify period for PrEP continuation (e.g., at month 1)

2. Discussion -The discussion could be strengthened by contextualizing the findings with the broader scientific literature.

3. Some statements appear too strong based on the data presented. For instance, "For example, among referred peers reached for follow-up, a greater percentage in the standard group reported an STI diagnosis (25%) and engagement in transactional sex (25%) in the past six months compared to those in the enhanced (9% and 13%, respectively)." These percentages are based on small absolute numbers; STI diagnosis is 5 (9%) and 7 (25%). I suggest deleting this statement as misleading when taking into consideration the small absolute number.

4. The author should expand on the limitations and also their potential effects on the estimates. Limitations of the study include a short follow-up period of only 1 month for PrEP refill (month 1), data was available only on 60% of the referred peers, and reliance on self-reported data, and most data were reported by index peers.

5. Lines 372- should add that the follow-up period includes only 1 PrEP refill (month 1). For example, "…and the relatively short follow-up period for all 372 study participants (three months covering month 1 PrEP refill only).

6. Line 341 is missing a number after "was".

---

* Please upload any figures associated with your paper as individual TIF or EPS files with 300dpi resolution at resubmission; please read our figure guidelines for more information on our requirements: http://journals.plos.org/plosmedicine/s/figures. While revising your submission, we strongly recommend that you use PLOS's NAAS tool (https://ngplosjournals.pagemajik.ai/artanalysis) to test your figure files. NAAS can convert your figure files to the TIFF file type and meet basic requirements (such as print size, resolution), or provide you with a report on issues that do not meet our requirements and that NAAS cannot fix.

After uploading your figures to PLOS's NAAS tool - https://ngplosjournals.pagemajik.ai/artanalysis, NAAS will process the files provided and display the results in the "Uploaded Files" section of the page as the processing is complete.

If the uploaded figures meet our requirements (or NAAS is able to fix the files to meet our requirements), the figure will be marked as "fixed" above. If NAAS is unable to fix the files, a red "failed" label will appear above.

When NAAS has confirmed that the figure files meet our requirements, please download the file via the download option, and include these NAAS processed figure files when submitting your revised manuscript.

FIGURES AND TABLES

SUPPLEMENTARY MATERIAL

REFERENCES

RCTs

* PLOS Medicine requires that all trials be prospectively registered in one of registries recognized by WHO. Please ensure that study registration details are included in the Methods section.

* Please structure the Methods section using the following sub-headings: Study design and participants, Randomization and masking, Procedures, Outcomes, Statistical analysis.

* The following outcomes measures [ADD DETAILS AS NEEDED OR DELETE BULLET POINT] appear to differ between the submitted manuscript and the protocol [and/or trial registry]. Please clarify and explain all discrepancies between the paper and protocol. If the outcomes were not prespecified in the protocol, please define them in the Methods (Outcomes section) as post hoc and explain why they were added. Post-hoc comparisons should be presented as hypothesis generating rather than conclusive.

* Please ensure that all prespecified outcomes (primary, secondary, and exploratory) are listed in the Methods/Outcomes section and indicate whether there are outcomes that are not presented in the current report.

* Please specify the dates (Month Day, Year) during which study enrollment and follow up occurred.

* Please include absolute numbers wherever you report percentages; eg, n/N (%)

* Please present the safety data for the study including numbers of specific events and whether or not adverse events are thought to be related to treatment. AEs should be reported in the abstract, per CONSORT and CONSORT-Harms.

* Please complete the CONSORT checklist (https://www.equator-network.org/reporting-guidelines/consort/) and ensure that all components of CONSORT are present in the manuscript, including how randomization was performed, allocation concealment, blinding of intervention, definition of lost to follow-up, power statement. When completing the checklist, please use section and paragraph numbers, rather than page numbers.

* Please report your abstract according to CONSORT for abstracts, following the PLOS Medicine abstract structure (Background, Methods and Findings, Conclusions) https://www.equator-network.org/reporting-guidelines/consort-abstracts/

* If your trial had to undergo important modifications in response to extenuating circumstances, please complete the CONSERVE-CONSORT checklist and provide in your Supporting Information; (https://www.equator-network.org/reporting-guidelines/guidelines-for-reporting-trial-protocols-and-completed-trials-modified-due-to-the-covid-19-pandemic-and-other-extenuating-circumstances-the-conserve-2021-statement/). When completing the checklist, please use section and paragraph numbers, rather than page numbers.

* In keeping with our commitment to Open Science, please include the study protocol document and analysis plan (including any amendments) as Supporting Information to be published with the manuscript if accepted.

* Please note that PLOS Medicine requires prospective, public registration of a data sharing plan (as part of mandatory clinical trials registration) for all clinical trials that began enrollment on or after January 1, 2019, in accordance with ICMJE requirements.

---

## [Decision Letter · Decision Letter 2]

19 Dec 2025

Dear Dr Ortblad,

Apologies for the delay in getting back to you with a decision which was due to challenges in securing all the necessary advice. Many thanks for re-submitting your manuscript "The effect of enhanced peer PrEP referral with HIV self-testing delivery among young Kenyan women: A randomized controlled trial of peer networks" (PMEDICINE-D-25-02363R2) to PLOS Medicine. The paper has been reviewed by the original Reviewer and a new Reviewer; their comments are included below and can also be accessed here: [LINK]

As you will see, while the original reviewers are not satisfied with the revisions, Reviewer 3 has raised some additional points. After discussing the paper with the editorial team and an academic editor with relevant expertise, I'm pleased to invite you to revise the paper in response to the Reviewer 3's comments.

We ask that you submit your revision by Jan 30 2026 11:59PM. However, if this deadline is not feasible, please contact me by email, and we can discuss a suitable alternative.

Don't hesitate to contact me directly with any questions (acunha@plos.org).

Best regards,

Andreia

Andreia Cunha, PhD

Senior editor

PLOS Medicine

acunha@plos.org

Comments from the reviewers:

Reviewer #1: Thank you to the authors for the response to my previous comments - their revision has addressed them well.

I did find it surprising to see the very large estimated ICC (0.66). After thinking about it, it makes sense if the number of referrals per peer was relatively low and the peer had a big influence (so that the rate of initiation was highly correlated within the cluster). I'll leave it to the authors if they wish to add more about this to the discussion point about the high ICC as it may be relevant to future work.

Reviewer #2: The authors have responded appropriately to the comments. I dont have any concerns with proceeding with the publication.

Reviewer #3: PMed review

This is a revision of a two-arm RCT examining PrEP uptake and HIV testing among adolescent and young girls in Kenya. Strengths include focusing on a group with an urgent need for HIV prevention, examining peer-based strategies, and a high follow-up rate among index participants. However, there were several notable weaknesses, including differential loss to follow up in the referred peer group, short follow-up period, and limited innovation. The intervention was expensive as well, raising some questions about what inferences can be made from this data in the current environment. While the revision was very responsive to comments, I have doubts about whether this is (yet) up to PLoS Medicine standards.

MAJOR COMMENTS

1) Innovation in the context of PrEP limited. There have been peer-based PrEP interventions before. There have also been many studies looking at peer-based HIVST interventions. In light of this research, what is new about this research? The innovation could be more clearly articulated.

2) Rationale for control arm unclear. The standard peer referral group would still need to have basic information about HIVST and the kits themselves. I am confused about the decision to not give informational brochures or kits to participants in this arm of the trial. A sentence explaining this more completely would be helpful.

MINOR COMMENTS

1) Line 37. "Inclusive of HIVST delivery" sounds a bit odd here. Consider instead: enhanced peer referral for HIVST and PrEP.

2) Lines 49-51 and 83-83. Not sure that I understand how finding that peer's motivated HIVST uptake has implications for nurses of chatbots being integrated into PrEP. Consider simplifying this around the data presented in both the highlights and abstract sections of the paper.

3) Line 53. There has been a pushback from African scholars against using SSA as a term. Would consider using "Africa" here, if possible and appropriate.

4) Line 58. RCT of peer networks is confusing here. Consider instead: " RCT of a peer-based intervention (NCTxx)…

5) Line 89. Can remove less than or equal to and define AGYW as simply "16 to 24 years"

6) Line 135. It might be easier here to refer to "alters" and "indexes" - using conventional network terms (and realizing that these will need to be defined for a PMed audience). Using the terms "index" and "index peers" is somewhat confusing. Also, the peers are sometimes called "index peers" and at other times called "referred peers." It is important to harmonize the nomenclature.

7) Line 150. Revise "competition" to "completion"

1) Line 425. There is also a literature showing that

8) Table titles. Recommend including the location, year, and sample size in each table title.

9) Figure nomenclature. What is called Figure 1 now is actually a table.

10) Protocol. The protocol attached in the supplement would benefit from having a section that lists changes. For example, it seems like there were relatively recent changes on NCT. An inventory of changes and their rationale would be useful here.

---

Editorial requests:

GENERAL

* Please review your text for claims of novelty or primacy (e.g. 'for the first time') and remove this language. In addition, please check that any use of statistical terms (such as trend or significant) are supported by the data, and if not please remove them.

* Statistical reporting: Please revise throughout the manuscript, including tables and figures.

- Please report statistical information as follows to improve clarity for the reader ""22% (95% CI [13,28]; p</=)"".

- Please separate upper and lower bounds with commas instead of hyphens as the latter can be confused with reporting of negative values.

- Please repeat statistical definitions (HR, CI etc.) for each set of parentheses."

* In the abstract, please include the important dependent variables that are adjusted for in the analyses.

FUNDING STATEMENT

* The funding statement should include: specific grant numbers, initials of authors who received each award, URLs to sponsors’ websites. Also, please state whether any sponsors or funders (other than the named authors) played any role in study design, data collection and analysis, the decision to publish, or preparation of the manuscript. If they had no role in the research, include this sentence: “The funders had no role in study design, data collection and analysis, decision to publish, or preparation of the manuscript.”

* It appears that one or more study authors is affiliated with one or more of the agencies that funded the study. Thus, the statement “The funders had no role in study design, data collection and analysis, decision to publish, or preparation of the manuscript” does not apply. Please revise the Financial Disclosure accordingly, as in "[Author name] is [author's role] at [funding agency]. The funders had no other role in study design…..”

COMPETING INTERESTS STATEMENT

* All authors must declare their relevant competing interests per the PLOS policy, which can be seen here: https://journals.plos.org/plosmedicine/s/competing-interests For authors with ties to industry, please indicate whether any of the interests has a financial stake in the results of the current study.

* Please add this statement to the manuscript's Competing Interests: "[Initials] is an Academic Editor on PLOS Medicine's editorial board."

DATA AVAILABILITY

* PLOS defines the “minimal data set” to consist of the data set used to reach the conclusions drawn in the manuscript with related metadata and methods, and any additional data required to replicate the reported study findings in their entirety. Authors do not need to submit their entire data set, or the raw data collected during an investigation. Please submit the following data:

The values behind the means, standard deviations and other measures reported;

The values used to build graphs;

The points extracted from images for analysis."

* The Data Availability Statement (DAS) requires revision. For each data source used in your study:

FIGURES

* Please provide titles and legends for all figures and tables (including those in Supporting Information files). Please define all acronyms used in each figure or table in its corresponding legend.

* Please ensure that where relevant figures include 95% CIs.

CLINICAL TRIALS

* Please complete the CONSORT 2025 checklist and ensure that all components of CONSORT 2025 are present in the manuscript, including how randomization was performed, allocation concealment, blinding of intervention, definition of lost to follow-up, power statement. When completing the checklist, please use section and paragraph numbers, rather than page numbers. The checklist should be included as supporting information, and should be cited in the article.

* PLOS Medicine requires that all trials be prospectively registered in one of registries recognized by WHO. Please provide information on study registration in the Methods section.

* In accordance with ICMJE requirements, PLOS Medicine requires prospective, public registration of a data sharing plan (as part of mandatory clinical trials registration) for all clinical trials that began enrollment on or after January 1, 2019.

* Please present the safety data for the study including numbers of specific events and whether or not adverse events are thought to be related to treatment.

* Causal language - In trials, there is usually a distinction in the language in terms of causal vs associational for primary and secondary trial outcomes. It would be beneficial to use associational language in the discussion and other sections for secondary outcomes.

* Please report your abstract according to CONSORT for abstracts, following the PLOS Medicine abstract structure (Background, Methods and Findings, Conclusions) https://www.equator-network.org/reporting-guidelines/consort-abstracts/

* Please include the clinical trial registry number in the abstract.

---

## [Decision Letter · Decision Letter 3]

21 Jan 2026

Dear Dr. Ortblad,

Thank you very much for re-submitting your manuscript "The effect of enhanced peer PrEP referral with HIV self-testing delivery among young Kenyan women: A randomized controlled trial of peer networks" (PMEDICINE-D-25-02363R3) for review by PLOS Medicine.

I have discussed the paper with my colleagues and the academic editor and it was also seen again by Reviewer 3. I am pleased to say that provided the remaining editorial and production issues are dealt with we are planning to accept the paper for publication in the journal.

[LINK]

We look forward to receiving the revised manuscript by Jan 28 2026 11:59PM.

Sincerely,

Andreia Cunha, PhD

Senior Editor

PLOS Medicine

plosmedicine.org

Comments from Reviewers:

Reviewer #3: Thanks to the authors for going the extra mile and ironing out the last few things. As a researcher I know that it's annoying when a new reviewer pops with different ideas, so kudos to the authors on this.

Requests from Editors:

GENERAL EDITORIAL REQUESTS

* At this stage, we ask that you include a short, non-technical Author Summary of your research to make findings accessible to a wide audience that includes both scientists and non-scientists. The Author Summary should immediately follow the Abstract in your revised manuscript. This text is subject to editorial change and should be distinct from the scientific abstract. Ideally each sub-heading should contain 2-3 single sentence, concise bullet points containing the most salient points from your study. In the final bullet point of ‘What Do These Findings Mean?’ Please include the main limitations of the study in non-technical language.

Please see our author guidelines for more information: https://journals.plos.org/plosmedicine/s/revising-your-manuscript#loc-author-summary."

* Please confirm that your title complies with PLOS Medicine's style. Your title must be nondeclarative and not a question. It should begin with main concept if possible. "Effect of" should be used only if causality can be inferred, i.e., for an RCT. Please place the study design ("A randomized controlled trial," "A retrospective study," "A modelling study," etc.) in the subtitle (ie, after a colon).

* Please ensure that the Introduction ends with a clear description of the study question or hypothesis.

* Please ensure that all abbreviations are defined at first use throughout the text.

* Please confirm that all numbers presented in the abstract are present and identical to numbers presented in the main manuscript text.

GENERAL

* Statistical reporting: Please revise throughout the manuscript, including tables and figures.

- Please report statistical information as follows to improve clarity for the reader (95% CI [13,28]; p</=).

- Please separate upper and lower bounds with commas instead of hyphens as the latter can be confused with reporting of negative values.

- Please repeat statistical definitions (HR, CI etc.) for each set of parentheses.

FUNDING STATEMENT

* The funding statement should include: specific grant numbers, initials of authors who received each award, URLs to sponsors’ websites. Also, please state whether any sponsors or funders (other than the named authors) played any role in study design, data collection and analysis, the decision to publish, or preparation of the manuscript. If they had no role in the research, include this sentence: “The funders had no role in study design, data collection and analysis, decision to publish, or preparation of the manuscript.”

COMPETING INTERESTS STATEMENT

* All authors must declare their relevant competing interests per the PLOS policy, which can be seen here: https://journals.plos.org/plosmedicine/s/competing-interests For authors with ties to industry, please indicate whether any of the interests has a financial stake in the results of the current study.

FIGURES

* Please ensure that where relevant figures include 95% CIs.

CLINICAL TRIALS

* Please complete the CONSORT 2025 checklist and ensure that all components of CONSORT 2025 are present in the manuscript, including how randomization was performed, allocation concealment, blinding of intervention, definition of lost to follow-up, power statement. When completing the checklist, please use section and paragraph numbers, rather than page numbers. The checklist should be included as supporting information, and should be cited in the article.

* Please report your abstract according to CONSORT for abstracts, following the PLOS Medicine abstract structure (Background, Methods and Findings, Conclusions) https://www.equator-network.org/reporting-guidelines/consort-abstracts/

* Please include the clinical trial registry number in the abstract.

* Please structure the Methods section using the following sub-headings: Study design and participants, Randomization and masking, Procedures, Outcomes, Statistical analysis.

* PLOS Medicine requires that all trials be prospectively registered in one of registries recognized by WHO. Please provide information on study registration in the Methods section.

* Please clarify if all deviations from the protocol or amendments have been approved and please describe these in the methods.

* Please add the inclusion/exclusion criteria that match the protocol to the Methods of the paper.

* In accordance with ICMJE requirements, PLOS Medicine requires prospective, public registration of a data sharing plan (as part of mandatory clinical trials registration) for all clinical trials that began enrollment on or after January 1, 2019.

* The primary, secondary and other outcomes measures do not appear to be present in the study protocol provided. Please clarify why this this is and which outcomes are being report in the manuscript, explaining any discrepancies between the paper and protocol.

* Please ensure that all prespecified outcomes (primary, secondary, and exploratory) are listed in the Methods/Outcomes section and indicate whether there are outcomes that are not presented in the current report.

* Please specify the dates (Month Day, Year) during which study enrollment and follow up occurred.

* Please also add exact dates of first and last patient enrolment.

* Please include absolute numbers wherever you report percentages; eg, n/N (%)

* Please present the safety data for the study including numbers of specific events and whether or not adverse events are thought to be related to treatment. AEs should be reported in the abstract, per CONSORT and CONSORT-Harms.

* In keeping with our commitment to Open Science, please include the study protocol document and analysis plan (including any amendments) as Supporting Information to be published with the manuscript if accepted.

[LINK]

---

## [Editor Report · Decision Letter 4]

12 Mar 2026

Dear Dr Ortblad,

On behalf of my colleagues and the Academic Editor, Dr Aaloke Mody, I am pleased to inform you that we have agreed to publish your manuscript "Effect of enhanced peer PrEP referral with HIV self-testing delivery among young Kenyan women: A randomized controlled trial of peer networks" (PMEDICINE-D-25-02363R4) in PLOS Medicine.

Before your manuscript can be formally accepted you will need to complete some formatting changes, which you will receive in a follow up email. Among other things they will ask you to please provide the exact dates of participant enrolment (day, month, yr, not just month, yr) in the following sentence in the Methods: 'From May 2023 to June 2024, 316 index peers were screened and 82 enrolled/randomized (median age 22 years, IQR 20-23): 40 to the enhanced group and 42 to the standard.' Please be aware that it may take several days for you to receive this email; during this time no action is required by you. Once you have received these formatting requests, please note that your manuscript will not be scheduled for publication until you have made the required changes.

PRESS

Sincerely,

Andreia Cunha, PhD

Senior Editor

PLOS Medicine